# Efficacy of combined administration of Baekhogainsam-Tang and low-dose pilocarpine on frequent intractable xerostomia: Study protocol for a randomized controlled trial

**Su Il Kim[1], Young Chan Lee[1], Ji Won Kim[2], Bo-Hyung Kim[3,4], Junhee Lee[5]\*, Young-Gyu Eun[6]\***

**1** Department of Otolaryngology-Head and Neck Surgery, Kyung Hee University School of Medicine, Kyung Hee University Hospital at Gangdong, Seoul, Korea, **2** Department of Otolaryngology-Head and Neck Surgery, Inha University College of Medicine, Incheon, Korea, **3** Department of Clinical Pharmacology and Therapeutics, Kyung Hee University Hospital, Seoul, Korea, **4** East-West Medical Research Institute, Kyung Hee University, Seoul, Korea, **5** Department of Sasang Constitutional Medicine, Kyung Hee University College of Korean Medicine, Kyung Hee University Korean Medicine Hospital, Seoul, Korea, **6** Department of Otolaryngology-Head and Neck Surgery, Kyung Hee University School of Medicine, Kyung Hee University Medical Center, Seoul, Korea

\* ygeun@khu.ac.kr (YGE); ssljh@khu.ac.kr (JL)

## Abstract

### Background

Intractable xerostomia is defined as the subjective perception of dry mouth and persistent salivary gland hypofunction. Pilocarpine is an approved salivary sialagogue that is frequently prescribed for the treatment of intractable xerostomia; however, it often exhibits more side effects at high-doses and limited effectiveness at low-doses. Baekhogainsam-Tang (BIT) is a common herbal formula used by patients complaining of sore throats and thirst. It seems that BIT can compensate for the insufficient effect of low-dose pilocarpine. However, no clinical trials have studied the efficacy of combined administration of BIT and low-dose pilocarpine for intractable xerostomia. We aim to assess the non-inferior efficacy and fewer side effects of combined administration of BIT and low-dose pilocarpine compared with the administration of high-dose pilocarpine.

### Methods

A randomized, open-label, parallel-group, multi-center trial will be conducted. A total of 120 patients with Sjogren's syndrome having an unstimulated salivary flow rate (SFR) $\leq 0.1$ mL/min or who have undergone radiotherapy to the head and neck with an unstimulated SFR $\leq 0.25$ mL/min will be recruited competitively. They will be randomly allocated to either the experimental or control groups. The experimental group will receive BIT herbal granules three times and pilocarpine (2.5-mg) four times daily; meanwhile, the control group will receive only 5-mg pilocarpine four times daily for 12 weeks. The primary outcome is

**Data Availability Statement:** Deidentified research data will be made publicly available when the study is completed and published.

**Funding:** SIK, YCL, JWK, BHK, JhL, YGE Grant numbers: HI20C1205 This work is supported by a grant from the Korea Health Technology R&D Project through the Korea Health Industry Development Institute (KHIDI), funded by the Ministry of Health and Welfare, Republic of Korea. URL: https://www.htdream.kr/ The funder had no role in the design, data collection, data analysis, and reporting of this study.

**Competing interests:** The authors have declared that no competing interests exist.

unstimulated SFR after 12 weeks of treatment. Secondary outcomes are stimulated SFR after 12 weeks of medication, as well as differences and mean percentage changes in unstimulated and stimulated SFR, visual analog scale, salivary scintigraphy, and questionnaires for both oral symptoms and quality of life during the clinical trial. An independent T test or Mann-Whitney U test will be performed to compare values between the two groups. The Paired T test or Wilcoxon signed-rank test will be performed to compare intragroup continuous values.

## Conclusion

This trial will be significant evidence on the efficacy and safety of combined use of BIT and low-dose pilocarpine to treat intractable xerostomia.

### Clinical trial registration

The Clinical Research Information Service of the Republic of Korea (ISRCTN, KCT0005982). Registered on 10 February 2021.

## Introduction

Xerostomia is defined as the subjective perception of dry mouth caused by various factors [1]. When the stimulated salivary flow rate is < 0.7 ml/min or the unstimulated salivary flow rate is < 0.1 ml/min, the condition is generally defined as salivary gland hypofunction. Intractable xerostomia is a condition in which dry mouth symptoms and salivary gland hypofunction persist even after treatment, including improvement in living conditions and stimulation of the salivary glands. The prevalence of xerostomia in patients with Sjogren's syndrome or those who have received radiotherapy for head and neck cancer is almost 100%, making them the representative causes of intractable xerostomia [2].

The treatment of intractable xerostomia requires addressing systemic diseases, adjusting medications and modifying lifestyle factors. Local spray medications, sugar-free gums, adequate hydration, and lubricants are commonly employed; however, these treatments have certain limitations. Pilocarpine has been approved by the US Food and Drug Administration (FDA) to increase salivary secretion in patients with Sjogren's syndrome or in those who have received radiotherapy for head and neck cancer [3]. Pilocarpine, an acetylcholine muscarinic M3 receptor agonist, functions as a sialagogue that stimulates the secretion of exocrine glands, including the parotid and submandibular glands [4]. However, its efficacy diminishes when a significant portion of the glands is damaged [5]. It often exhibits more side effects at high-doses and limited effectiveness at low-doses. Prolonged high-dose pilocarpine use may lead to various side effects, including sweating, chills, nausea, dizziness, increased urinary frequency, and palpitations [6]. Moreover, the use of pilocarpine is contraindicated in patients with severe asthma, chronic obstructive pulmonary disease, glaucoma, iritis, and taking beta-blockers.

Baekhogainsam-Tang (BIT; Baihu Jia Renshen-tang in Chinese; Byakko-ka-ninjin-to in Japanese) is known to be effective in treating symptoms such as flushing, thirst, and sweating due to dehydration during the chronic stage of high fever [7, 8]. BIT, widely used in Korea, China, and Japan, has been approved by the Korea FDA for clinical use in patients complaining of sore throat and thirst. In two previous animal experiments, BIT increased salivary gland secretion [7] and improved the effect of anticholinergic agents on salivary secretion but had no

influence on the bladder [9]. In a study that examined toxicity by administering BIT (2000 mg/kg/day in a laboratory setting) in rats for 13 weeks, no adverse reaction related to death or medication were observed in the administration of significant amount of BIT [10]. In other words, when pilocarpine and BIT were administered in combination in mice, BIT increased the salivary secretion of pilocarpine without significant side effects. In a previous study in human, only BIT was administered in 30 elderly patients with xerostomia, of which 60% reported an improvement in symptoms [11]. Thus, it is thought that BIT can compensate for the insufficient effect of low-dose pilocarpine. However, to the best of our knowledge, no clinical trials have studied the efficacy of combined administration of BIT and low-dose pilocarpine for human patients with intractable xerostomia, such as Sjogren's syndrome, or in those who received radiotherapy for head and neck cancer.

We believed that, similar to pilocarpine, BIT would effectively relieve symptoms in patients with intractable xerostomia. Additionally, we hypothesized that patients with intractable xerostomia taking both BIT and low-dose pilocarpine would have similar symptom relief but fewer side effects compared with those taking only high-dose pilocarpine.

The aim of this prospective, randomized, open-label, parallel-group, multi-center, and controlled trial is to ascertain whether combined administration of BIT herbal medicine and low-dose pilocarpine can demonstrate both non-inferior efficacy and fewer side effects in patients with frequent intractable xerostomia (patients who underwent radiation therapy to the head and neck or with Sjogren's syndrome) than the administration of only high-dose pilocarpine.

## Materials and methods

### Trial design

This study is a prospective, randomized, open-label, parallel-group, multi-center, and controlled trial involving participants from four clinical research centers in Korea: Kyung Hee University Medical Center, Kyung Hee University Hospital in Gangdong, Inha University Hospital, and Myongji Hospital. This trial was registered with the Clinical Research Information Service of the Republic of Korea (ISRCTN, KCT0005982, Registered 10 February 2021). The current version of the protocol is version 2.6, dated 14, November 2023. First participant recruitment commenced on 5 August 2021. It is anticipated that recruitment and analysis of the trial data will be completed by December 2024.

The summary schedule of enrolment, interventions, and assessments is shown in Fig 1, and the flowchart of the study is shown in Fig 2. Eligible participants will be randomly allocated to either the experimental or control group after screening based on the inclusion and exclusion criteria. The patients will receive treatment for 12 weeks. The unstimulated and stimulated salivary flow rate (SFR) test, visual analog scale (VAS), salivary scintigraphy, and questionnaires for oral symptoms and quality of life will be assessed for up to 12 weeks after the first visit.

### Participants and eligibility

**Inclusion and exclusion criteria.** The inclusion criteria are as follows: (1) Patients with healthy oral mucosa who undergone radiation therapy to the head and neck or those with primary Sjogren's syndrome (positive anti-Ro/SS-A antibody reaction and/or labial salivary gland biopsy showing focal lymphocytic sialadenitis and focus score of $\geq 1$ foci/4 mm$^2$, and unstimulated SFR $\leq 0.1$ ml/min) [12]; (2) unstimulated SFR $\leq 0.25$ ml/min [13]; (3) VAS score for dry mouth over the past one month $\geq 4$; (4) age >19 years; (5) compliance with all written informed consent.

Patients with the following past medical history are excluded [14, 15]: heart failure, medication-refractory hypertension (systolic blood pressure $\geq 160$ mmHg or diastolic blood

| | STUDY PERIOD | | | | |
|---|---|---|---|---|---|
| | Enrolment | Allocation | Post-allocation (treatment) | | |
| TIMEPOINT** | -t1<br>(-1w~0d) | t0<br>(0d) | t1<br>(4w±2d) | t2<br>(8w±2d) | t3<br>(12w±2d) |
| ENROLMENT: | | | | | |
| Eligibility screen | X | | | | |
| Informed consent | X | | | | |
| Allocation | | X | | | |
| INTERVENTIONS: | | | | | |
| Experimental group (BIT + 2.5 mg pilocarpine) | | | ●————————————● | | |
| Control group (5 mg pilocarpine) | | | ●————————————● | | |
| ASSESSMENTS: | | | | | |
| Physical examination | X | X | X | X | X |
| Vital signs | X | X | X | X | X |
| Blood tests | X | | X | X | X |
| OHIP-14 | | X | X | X | X |
| EQ-5D-5L | | X | X | X | X |
| Visual analogue scale (VAS) | X | | X | X | X |
| Salivary flow rate (SFR) test | X | | X | X | X |
| Salivary scintigraphy | | X | | | X |
| Monitoring adverse events | | X | X | X | X |
| Evaluating concomitant medication | | X | X | X | X |

**Fig 1. Summary schedule of enrolment, interventions, and assessments.** OHIP-14, The 14-item Oral Health Impact Profile; EQ-5D-5L, the 5-level EuroQol 5-dimensional questionnaire.

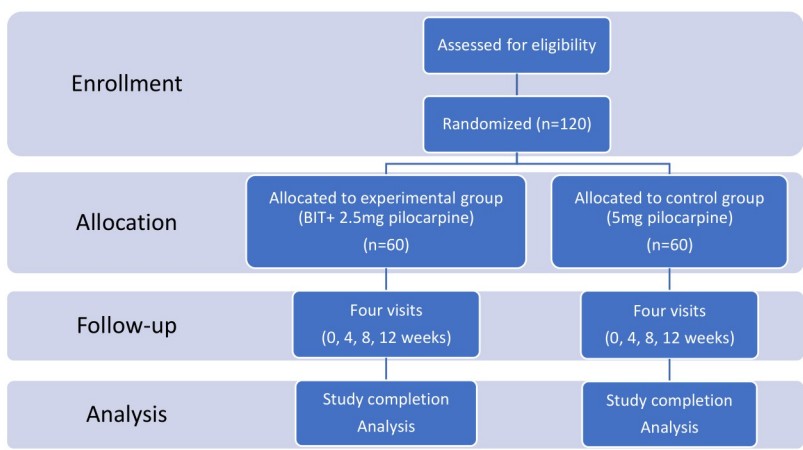

**Fig 2. Flow chart of the study.**

pressure $\geq$ 100 mmHg), bronchial asthma, arrhythmia accompanied by clinical symptoms or requiring treatments, coronary artery disease, chronic obstructive pulmonary diseases, chronic bronchitis, inborn errors of amino acid metabolism, hepatic encephalopathy, ophthalmic diseases (such as narrow angle glaucoma, peripheral retinopathy, and iritis). Patients taking following medications are also excluded. These are prohibited drug in combination with the trial drugs or drugs causing xerostomia: coumadin, heparin, warfarin, aspirin, anticholinergics, first-generation antihistamines, antidepressants with anticholinergic effects, monoamine oxidase inhibitors, diuretics, mineralocorticoids, antiepileptics, opioid analgesics, β-adrenaline antagonist, or digitalis; patients taking low-dose aspirin or β-adrenaline antagonist can participate in the protocol based on the cardiologist's opinion.

Additionally, we exclude patients with clinical laboratory test results falling within the following ranges: hemoglobin < 8.0 g/dL, aspartate transaminase (AST) or alanine transaminase (ALT) > 3 times the upper limit of each hospital standard, total bilirubin > 2 times the upper limit of each hospital standard, creatinine >1.5 times the upper limit of each hospital standard, or creatinine clearance ≤ 60 mL/min. Other exclusion criteria are as follows [16]: (1) pregnancy or lactation; (2) disagreement with proper contraception; (3) galactose intolerance, Lapp lactase deficiency, or glucose-galactose malabsorption; (4) difficulty in oral administration of a clinical trial drug; (5) significant hypersensitivity to BIT or pilocarpine; (6) cognitive disorder; and (7) inability to read and write.

**Recruitment.** Patients presenting with dry mouth symptoms at the otolaryngology clinic of each medical center will be prospectively investigated. A total of 120 patients will be competitively recruited from the four trial centers.

**Randomization, allocation concealment, and blinding.** Random numbers will be generated by a computerized random-number generator using the block-randomization method of SAS version 9.1.3 (SAS Institute Inc., NC, Cary, USA) for sequence generation. An independent statistician, blinded to the study design and purpose, will perform the random allocation of a total of 120 patients to the experimental group (BIT + 2.5 mg pilocarpine) or the control group (5 mg pilocarpine) in a 1:1 ratio using a stratified block randomization method (a block size of 4). The statistician will create opaque blind assignment envelopes (with consecutive numbers) and deliver them to each participating center.

Participants who pass the screening test during their first visit (screening: 1 week) will be assigned a registration number during the second visit (baseline trial week). Based on the registration number assigned to each participant, the corresponding envelope will be opened in front of the participant to reveal the treatment methods. Either BIT + 2.5 mg pilocarpine or 5 mg pilocarpine treatments will be initiated after random allocation. Because the drugs are identifiable, clinicians and patients will be aware of the treatment group assignments after random allocation.

## Interventions

**Experimental group (BIT + 2.5 mg pilocarpine) and control group (5 mg pilocarpine).** The experimental group will receive both BIT herbal granules and 2.5 mg pilocarpine (half the effective dose) [17]. BIT granules will be taken orally before each meal three times, and pilocarpine (2.5 mg) will be administered four times (after each meal and before bedtime) daily for 12 weeks.

The components of the BIT granules (4 g / dose) are presented in Table 1. Each dose of four grams (dry weight) of granules contain five herbs: *Anemarrhena Rhizome*, *Gypsum*, *Licorice*, *Oryzae Semen*, *and Ginseng*. This 11,290 mg mixture yields 830 mg of soft extract after boiling in water, filtering, and concentrating. Following the addition of one diluent (Lactose Hydrate)

**Table 1. Components of Baekhogainsam-Tang (BIT) granules (4 g / dose) [18].**

| Scientific name | Application category | Amount (mg) |
|---|---|---|
| *Anemarrhena Rhizome* | Main component | 2,000 |
| *Gypsum* | Main component | 4,330 |
| *Licorice* | Main component | 660 |
| *Oryzae Semen* | Main component | 3,300 |
| *Ginseng* | Main component | 1,000 |
| *Lactose Hydrate* | Diluent | 2,260 |
| *Corn Starch* | Excipients | q.s. |
| *Hydroxypropyl Cellulose* | Excipients | q.s. |
| *Sucrose Fatty Acid Ester* | Excipients | q.s. |
| *Magnesium Stearate* | Excipients | q.s. |
| *Light Anhydrous Silicate* | Excipients | q.s. |
| Total | | 4,000 |

q.s.; Quantum sufficit

and five excipients, 4 g of BIT is obtained. The BIT granules were purchased from Hanpoong Pharmaceutical & Food Co., Ltd. (Jeonju, Republic of Korea). Voucher specimens will be reserved in the research library of Hanpoong Pharm & Food Company.

To minimize the side effects that can occur in the control group, which will receive 5 mg of pilocarpine [19], only 2.5 mg will be prescribed to the experimental group. After the screening test (visit 1), the patients will visit four times over 12 weeks (baseline, week 4, week 8, and week 12). The window for visits is ±2 days during the visit periods. Visits one and two can be performed simultaneously. Details of the study protocol are summarized in Fig 1 according to the Standard Protocol Items: Recommendations for Interventional Trials (SPIRIT) guidance [20]. The completed SPIRIT checklist is available as a S1 Appendix.

The control group will receive 5 mg of pilocarpine four times (after each meal and before bedtime) daily for 12 weeks. Because these patients do not receive BIT granules, upon receiving the medication, the patients in both groups will be aware of their group assignment (open-label). The patients in the control group will visit the outpatient clinic in the same manner as those in the experimental group.

## Outcomes

**Primary outcome.** The primary outcome of this study is the unstimulated SFR after 12 weeks of treatment, which will be presented as mean ± standard deviation (SD) in each group, respectively. To determine the unstimulated SFR, the participants will abstain from food intake for at least one hour. The participants will need to rinse their mouth with water, held it open, and gather saliva in a paper cup for 5 min. Then, the saliva collected in the paper cup was placed in the graduated test tube to accurately measure the amount of saliva. SFR (ml/min) is calculated by dividing the measured amount of saliva (ml) by time (5 min).

**Secondary outcomes.** The secondary outcomes include the following: (1) Stimulated SFR after 12 weeks of medication. (2) Differences and mean percentage changes in unstimulated and stimulated SFR from baseline (week 0) to the end of the trial (week 12). (3) Differences and mean percentage change in the VAS, 14-item Oral Health Impact Profile (OHIP-14), and the 5-level EuroQol 5-dimensional (EQ-5D-5L) questionnaire from baseline (week 0) to the end of the trial (week 12). (4) Differences and mean percentage changes in the uptake ratio

(UR), maximum accumulation (MA), and maximum secretion (MS) on salivary scintigraphy from baseline (week 0) to the end of the trial (week 12). Examination at baseline (week 0) can be replaced by examination at the time of screening (week 1).

The stimulated and unstimulated SFR will be assessed at every visit from baseline (week 0) to 12 weeks (weeks 4, 8, and 12). To determine the stimulated SFR, the participants will be encouraged to suck sugar-free lemon candy. The procedure for measuring the stimulated salivary flow is the same as that for measuring the unstimulated salivary flow.

The degree of symptoms associated with dry mouth and oral pain will be assessed using the VAS (no symptoms to maximum symptoms; score, 0–10). Subjective quality of life will be evaluated using OHIP-14 Korean version [21, 22]. The responses will be measured on a scale ranging from 0 (never) to 4 (always), generating a total score of 0 to 56 (lowest quality). The patients' health status will be evaluated using the EQ-5D-5L questionnaire [23]. This questionnaire consists of five dimensions (mobility, self-care, usual activities, pain/discomfort, and anxiety/depression), each with five levels (no problems, slight problems, moderate problems, severe problems, and extreme problems; scores of 1 to 5). The EQ-5D-5L health states are defined by combining one level from each of the five dimensions, for a total of 3125 possible health states. The VAS, OHIP-14, and EQ-5D-5L will be checked at every visit from baseline (week 0) to 12 weeks (weeks 4, 8, and 12).

For the objective evaluation of salivary gland function, salivary scintigraphy will be performed after the administration of a radiotracer (Technium-99m pertechnetate). The detailed process is described in a previous study [22]. To analyze the scintigraphy images, the counts of the regions of interest (ROI) will be evaluated in the parotid and submandibular glands. Three parameters–UR, MA, and MS–will be calculated using the following formulas: (1) UR is calculated as the salivary gland count–the background count. (2) The MA (%) is calculated as follows: (maximum activity before stimulation–initial activity 5 min after radiotracer injection) × 100 / maximum activity before stimulation. (3) The MS (%) is calculated as (maximum activity prior to stimulation–activity after stimulation) × 100 / peak activity before stimulation. Salivary scintigraphy will be performed at baseline (week 0) and at the final visit (week 12).

**Adverse events reporting.** An adverse event (AE) refers to an undesirable and unintended sign, symptom, or disease that occurs in a patient who received clinical trial drugs but does not necessarily have a causal relationship with the clinical trial drugs. Each patient will be monitored for AE after each visit, and appropriate measures will be taken immediately if necessary. The severity of AE will be assessed in accordance with the MedDRA version 23.1. All AE will be recorded in the patients' medical records and electronic case report forms (e-CRFs).

**Safety monitoring.** Safety will be investigated using AE reporting, vital signs, physical examinations, and clinical laboratory tests. Vital signs and physical examinations will be checked at all visits throughout the 12 weeks (visits 1–5). Clinical laboratory tests will be performed at every visit from baseline (week 0) to 12 weeks (weeks 4, 8, and 12). These tests include hemoglobin, white blood cell, platelet count, AST, ALT, total bilirubin, creatinine, serum electrolyte, and urinary tests.

**Withdrawal, dropout, and discontinuation.** Patients can withdraw their consent to participate in the clinical trial at any time and discontinue their participation. Additionally, clinical trials will be suspended under the following circumstances: (1) Participants' conditions that fit the exclusion criteria of the clinical trial selection are determined after screening. (2) A new violation of the clinical trial is discovered. (3) Significant AE or drug reactions. (4) Participants took a prohibited drug in combination during the clinical trial. (5) Where the progress of the clinical trial is deemed inappropriate based on the judgement of the principal or sub-investigator.

**Data management and trial monitoring.** All procedures will be delivered via face-to-face individual interview and managed using REDCap electronic data collection software by trained research assistants. The data monitoring committee (DMC) composed of principal investigators, co-investigators, and research assistants will regularly monitor the implementation of study procedures in each trial center. This committee is independent from the sponsor and competing interests. All AEs will be reported to the Korea FDA and IRB committee by the DMC.

## Sample size

The primary outcome of this study is the unstimulated SFR after treatment for 12 weeks The sample size for this study was calculated based on the results of a previous study [17]. In the previous study, the mean ± SD of unstimulated SFR for 5 mg pilocarpine therapy vs placebo group after 12 weeks of medication, respectively, were 0.38 ± 0.48 vs 0.17 ± 0.13 mL/min (p <0.001). Based on these results, the SD was calculated to be 0.35, and the non-inferiority limit was calculated as 0.21 (point estimation; 0.38−0.17 = 0.21). The SD ($\sigma$) of the population was calculated by dividing 0.35 by 0.21 (non-inferiority limit). Considering a one-tailed 5% significance level ($\alpha = 0.05$), 90% power ($\beta = 0.1$), and 1:1 ratio of experimental and control groups, approximately 48 participants were calculated to be required in each group. Assuming a 20% dropout rate, we calculated the sample size as 120 participants (60 in each group).

## Statistical methods

**Patient group for data analysis.** The intention-to-treat (ITT) population consists of participants who received at least one dose of the clinical trial drugs and underwent efficacy evaluation more than once during the treatment period. In cases of missing values due to dropouts or discontinuation before the clinical trial ends, missing data from dropout participants will be imputed using the last observation carried forward analysis.

The per-protocol (PP) population is a subset of the ITT population. It includes participants who consumed >70% of the prescribed doses of clinical study drugs and completed follow-up visits and corresponding outcome measurements (weeks 4, 8, and 12) during the 12-weeks treatment periods. Participants falling under a 'significant violation of clinical trial plan' will be excluded from the PP population.

Statistical analyses will be conducted on both the ITT and PP populations based on the evaluation values. Demographic information will be assessed for all participants. Safety outcomes will be analyzed in the ITT population. Efficacy outcomes will be analyzed for both the PP and ITT populations. In case of discrepancies between the efficacy outcomes of PP and ITT populations, both set of results will be presented and compared, with the PP population considered the main analysis and the ITT population as supplementary analysis.

**Statistical analysis.** Data will be presented as the mean ± SD for continuous data or frequencies for categorical data. To confirm the validity of the random allocation, demographic and baseline values between the experimental and control groups will be compared and evaluated. We will use the independent t-test or Mann-Whitney U test for continuous outcome measures and the chi-square test, Fisher's exact test, or Cochran-Mantel-Haenszel Method for categorical outcome measures.

A 95% confidence interval (95% CI) will be used to analyze primary outcomes. We will calculate the 95% CI of the mean difference in the unstimulated SFR between the two groups after 12 weeks of medication. If the lower end of the 95% CI of the mean difference is greater than the non-inferiority limit (−0.21), it will be concluded that the experimental group is not inferior to the control group. For further analysis, if the SFR between the two groups satisfy

the normality test, a t-test will be performed to compare the two groups in terms of the primary outcomes; otherwise, the Mann-Whitney U test will be performed. If there is a significant difference in baseline values between the two groups, repeated-measures analysis of covariance (ANCOVA) will be performed.

To analyze the secondary outcomes, in the case of continuous outcome measures, a normality test for the distribution of data will be initially performed. In the case of non-normally distributed data, the data will be transformed to a normal distribution using the log-transformation or square root transformation methods. The analysis will be proceeded using either parametric or non-parametric methods. Specifically, an independent T test or Mann-Whitney U test will be performed to compare values between the two groups. The Paired T test or Wilcoxon signed-rank test will be performed to compare intragroup continuous values. The group and time interaction effect of repeated-measures data will be analyzed through repeated-measures analysis of variance (ANOVA) or ANCOVA, or through the GEE model test based on the nature of the data. The level of significance will be set at $p < 0.05$.

## Ethics and dissemination

**Ethics.** The study protocol was approved by the Institutional Review Board (IRB) of Kyung Hee University Medical Center (approval no. 2020-10-032), Kyung Hee University Hospital at Gangdong (2022-03-020), Inha University Hospital (2020-09-012), and Myongji Hospital (2020-09-014) before participants enrolment.

All participants will be provided with information regarding the study protocol, and written informed consent will be obtained from all eligible participants before enrolment.

**Trial status.** This clinical trial has been registered with the Clinical Research Information Service the Republic of Korea (ISRCTN, KCT0005982) on 10 February 2021. Participant recruitment has commenced after that. The date of first participant enrolment was 5 August 2021. The research plan was revised to protocol version 2.6 as one researcher of Myongji Hospital retired and moved to Kyung Hee University Hospital at Gangdong. This trial is anticipated to be completed by December 2024.

**Dissemination.** Access to the e-CRF is restricted by user password and role permission access. Each user will be assigned to a pre-defined role such as principal investigator, sub-investigator, and data manager. A list of people who have been authorized to modify the data was described and kept. The completed original e-CRF is the exclusive property of the client, and may not be disclosed to a third party without written approval from the client.

The results of the clinical trial will be disseminated to the clinicians dealing with xerostomia. This process will be implemented on a society level (publication, public talks, news) and scientific level (peer-reviewed journals, presentations at international oral disease society).

## Discussion

Intractable xerostomia causes persistent discomfort despite continuous treatment; however, a standardized treatment protocol has not been established yet. Pilocarpine, a sialagogue approved by the US FDA, is less effective when there are few residual salivary glands and is accompanied by many contraindications and side effects. Mild muscarinic side effects, including sweating, urinary frequency, flushing, nausea, and dizziness, seem to be common with therapeutic use and are dose-dependent [24]. Administration of low-dose pilocarpine (2.5 mg) can reduce side effects but also diminishes its effectiveness as a sialagogue [17].

BIT is an herbal medicine that has long been used to treat sore throat, thirst, and flushing; thus, it may be added as an adjuvant therapy for intractable xerostomia. However, limited research exists on its efficacy in xerostomia, especially in cases of intractable xerostomia

derived from Sjögren's syndrome or radiation therapy for head and neck cancer. Therefore, we aim to demonstrate the efficacy and safety of combined administration of BIT herbal medicine and low-dose pilocarpine compared with the administration of only high-dose pilocarpine for intractable xerostomia.

In a previous study using mice, it was observed that *Anemarrhena Rhizome* and *Gypsum* (calcium sulfate) components of BIT interact to increase the salivary secretion of pilocarpine [25]. When timosaponin-AIII, the constituents of *Anemarrhena Rhizome*, was administered to mice, it showed about 175-fold more potent on salivary secretion than the only BIT extracts. Especially, the effect of timosaponin-AIII on salivary secretion was strengthened when *Gypsum* was added together, which is thought to be because the secretion of salivary gland is influenced by the intracellular calcium concentration in the acinar cells of the salivary gland.

This clinical trial was designed in accordance with the SPIRIT statement [20]. This trial has been designed as a randomized, open-label, parallel-group, multi-center study. Although blinding is an important method in randomized clinical trials, clinicians and patients are aware of their group (open-label) assignments due to identifiable drugs. Open-label studies have a potential reporting bias for patient-reported outcomes such as VAS, OHIP-14, and EQ-5D-5L. However, objective outcomes, such as SFR and scintigraphy, are not affected by these factors. Additionally, to avoid selection bias in this open-label study setting, the patients will be randomly assigned across multiple centers.

Our study has some limitations. First, there was no placebo group in this study. The aim of this study was to ascertain whether the experimental group (combined administration of BIT and low-dose pilocarpine) can demonstrate both non-inferior efficacy and fewer side effects than the control group (administration of only high-dose pilocarpine). We planned a blinding trial and tried to make placebo or mimics for BIT and pilocarpine. However, considering the characteristics of BIT granules, it was very difficult to make it with the same formulation as pilocarpine. Additionally, half dose pilocarpine should also be manufactured as a capsule due to the case of taking half dose pilocarpine in experimental group, which was also difficult to the nature of pilocarpine. Thus, the trial was unblinded. Second, patient recruitment was not as smooth as expected, targeting only patients who underwent radiation therapy to the head and neck or with primary Sjogren's syndrome among patients with xerostomia. Since the recruitment of patients with secondary Sjogren's syndrome is likely to involve more subjective factors than primary Sjogren's syndrome, we planned to recruit and analyze only patients with objective diagnostic criteria in this study. We plan to recruit and analyze comprehensive patients with xerostomia from more clinical centers in subsequent clinical trials based on the results of this study.

In conclusion, this trial will provide essential evidence on the efficacy and safety of combined use of BIT and low-dose pilocarpine in treating intractable xerostomia. Based on the results of this clinical trial, a future larger-scale clinical trial evaluating the effectiveness and safety of BIT in treating intractable xerostomia derived from various causes will be feasible.

## Supporting information

**S1 Appendix. SPIRIT checklist of this study.**
(DOCX)

**S2 Appendix. Detailed trial study protocol (original language).**
(PDF)

**S3 Appendix. Detailed trial study protocol (English translation).**
(PDF)

## Author Contributions

**Conceptualization:** Su Il Kim, Junhee Lee, Young-Gyu Eun.

**Data curation:** Su Il Kim, Young Chan Lee, Ji Won Kim, Bo-Hyung Kim, Junhee Lee, Young-Gyu Eun.

**Formal analysis:** Young Chan Lee, Bo-Hyung Kim, Junhee Lee, Young-Gyu Eun.

**Funding acquisition:** Young-Gyu Eun.

**Investigation:** Su Il Kim, Young Chan Lee, Ji Won Kim, Bo-Hyung Kim, Junhee Lee.

**Methodology:** Bo-Hyung Kim, Junhee Lee.

**Project administration:** Young-Gyu Eun.

**Resources:** Young-Gyu Eun.

**Software:** Bo-Hyung Kim, Junhee Lee.

**Supervision:** Junhee Lee, Young-Gyu Eun.

**Visualization:** Su Il Kim, Young Chan Lee, Ji Won Kim, Bo-Hyung Kim, Junhee Lee, Young-Gyu Eun.

**Writing – original draft:** Su Il Kim, Junhee Lee, Young-Gyu Eun.

**Writing – review & editing:** Su Il Kim, Young Chan Lee, Ji Won Kim, Bo-Hyung Kim, Junhee Lee, Young-Gyu Eun.

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
