## [Decision Letter · Decision Letter 0]

5 Jun 2024

PONE-D-24-16207Efficacy of combined administration of Baekhogainsam-Tang and low-dose pilocarpine on frequent intractable xerostomia: Study protocol for a randomized controlled trialPLOS ONE

Dear Dr.  Eun,

Thank you for submitting your manuscript to PLOS ONE. After careful consideration, we feel that it has merit but does not fully meet PLOS ONE’s publication criteria as it currently stands. Therefore, we invite you to submit a revised version of the manuscript that addresses the points raised during the review process.

We look forward to receiving your revised manuscript.

Kind regards,

Hadi Ghasemi

Academic Editor

PLOS ONE

Journal Requirements:

Reviewers' comments:

Reviewer's Responses to Questions

**Comments to the Author**

1. Does the manuscript provide a valid rationale for the proposed study, with clearly identified and justified research questions?

Reviewer #1: Yes

Reviewer #2: Yes

Reviewer #3: Yes

Reviewer #4: Yes

2. Is the protocol technically sound and planned in a manner that will lead to a meaningful outcome and allow testing the stated hypotheses?

Reviewer #1: Yes

Reviewer #2: Yes

Reviewer #3: Yes

Reviewer #4: Yes

3. Is the methodology feasible and described in sufficient detail to allow the work to be replicable?

Reviewer #1: Yes

Reviewer #2: Yes

Reviewer #3: Yes

Reviewer #4: Yes

4. Have the authors described where all data underlying the findings will be made available when the study is complete?

Reviewer #1: Yes

Reviewer #2: Yes

Reviewer #3: Yes

Reviewer #4: No

5. Is the manuscript presented in an intelligible fashion and written in standard English?

Reviewer #1: Yes

Reviewer #2: Yes

Reviewer #3: Yes

Reviewer #4: Yes

6. Review Comments to the Author

You may also provide optional suggestions and comments to authors that they might find helpful in planning their study.

Reviewer #1: Appears to be a well-conceived protocol. Would the authors consider adding Cevimeline as an additional experimental group?

Reviewer #2: well written and much needed research on the topic.The methodology is described in detail and can be replicated easily.All relevant outcomes are measured.

Reviewer #3: Dear Authors,

interesting study protocol for stubborn xerostomia.

In exclusion criteria you didn't mentioned use of antiepileptics and opioid analgesics such as tramadol which are strongly xerogenic drugs.

It is also very important to include only patients with healthy oral mucosa, without any evidence of infection such as oral candidiasis, which is very often in these patients and may compromise the effect of the applied therapy.

In outcomes section please explain more precisely how saliva will be collected and measured? Will it be with funnel and graduated test tube or with something else?

Discussion needs to be expanded.

Best regards

Reviewer #4: I appreciate the opportunity to review this interesting study protocol on the Efficacy of combined administration of Baekhogainsam-Tang and low-dose pilocarpine on frequent intractable xerostomia. I enjoyed reading the manuscript. I commend the authors for a number of strengths of their work, including:

1- Simultaneous quantitative (salivary amount) and qualitative investigations in the field of xerostomia in this study can achieve more reliable results.

2- A large sample size that will be taken from several medical center will strengthen the diversity of the sample.

3- The methodology is written in good details that the reproducibility of the study can be achieved.

Considering these strengths, though, as I read the manuscript I found some areas in which I would have appreciated greater clarity. I believe the paper could be further strengthened by considering the follow items:

Abstract:

1- Please mention the clinical trial registration and add a summary of the statistical analysis method to the methods part. Also, please include conclusion in the abstract.

2- Since another muscarinic agent (cevimeline), have recently been approved for the treatment of dry mouth in Sjögren syndrome and ...), please revise the sentence of "Pilocarpine is a solely approved salivary sialagogue to treat intractable xerostomia" to (it seems pilocarpine is best performing sialogogue drug for treatment of intractable xerostomia)

3- Please write keywords in alphabetical order according to MESH.

Introduction:

1- In line 86 when you explain about BTT, please mention the side effects and toxic dose of BTT (if possible).

Material and methods:

1- In trial design section: The status and timeline of the study is not mentioned. (Please add how long will the overall process of the study take from the time of the initial screening of people until the results of the study are obtained?)

2- In inclusion and exclusion criteria section:

a. Since the users of a wide range of medicines have been mentioned as part of your exclusion criteria, and patients with secondary Sjogren's syn. require the use of a wide range of mentioned medicines due to the simultaneous suffering of other connective tissue diseases (such as SLE or RA), Can it be said that the primary Sjogren's syn. is your inclusion criteria? (Just Primary Sjogren instead of Sjogren). In that case, can minor salivary gland biopsy be added to this part (inclusion criteria) as the gold standard criteria for primary Sjogren's syn. diagnosis?

b. Please add references to exclusion criteria.

3- In Interventions section:

a. Considering that you have mentioned in the introduction of the article that "no clinical trials have studied the efficacy of combined administration of BTT and low-dose pilocarpine for intractable xerostomia", is there any previous study on possible medical interactions or anti-synergism effects of BTT and pilocarpine (even as invitro studies)? (please mention)

b. Please include a reference for exact amounts of BTT components.

Discussion:

1- Please add the limitations of the study design

2- Please add the interaction mechanism between the BTT components and to explain the combined effects of that components with pilocarpine (for example: interaction between some Anemarrhena components and CaCl2 is a way to explain the combined effects of BTT with pilocarpine and ....)

7. PLOS authors have the option to publish the peer review history of their article (what does this mean?). If published, this will include your full peer review and any attached files.

Reviewer #1: **Yes: **Elliot Abt

Reviewer #2: No

Reviewer #3: No

Reviewer #4: No

---

## [Author Response · Author response to Decision Letter 0]

11 Jun 2024

Dear Dr Hadi Ghasemi

Academic Editor

PLOS ONE

We wish to re-submit a revised manuscript titled “Efficacy of combined administration of Baekhogainsam-Tang and low-dose pilocarpine on frequent intractable xerostomia: Study protocol for a randomized controlled trial” by Su Il Kim et al., for consideration as a publication in the PLOS ONE. The manuscript number is PONE-D-24-16207.

We would like to thank editor and reviewers for their helpful comments and suggestions. We have revised our manuscript as per these comments. The following is a point-by-point response to the suggestions (bold) of the editor and reviewers (responses to the comments are provided in blue-colored text). The revised text in the manuscript has been highlighted in yellow. We also have revised our initial manuscript’s format according to the PLOS ONE’s style requirements. Additionally, figure files (fig. 1 and 2) were uploaded to PACE digital diagnostic tool and adjusted. (Resolution change to 300 PPI & TIFF file conversion to valid TIF file format).

Thank you very much for your consideration of our manuscript. We look forward to hearing from you.

Sincerely,

Young-Gyu Eun, MD, PhD

---

## [Decision Letter · Decision Letter 1]

21 Jun 2024

PONE-D-24-16207R1Efficacy of combined administration of Baekhogainsam-Tang and low-dose pilocarpine on frequent intractable xerostomia: Study protocol for a randomized controlled trialPLOS ONE

Dear Dr. Eun,

Thank you for submitting your manuscript to PLOS ONE. After careful consideration, we feel that it has merit but does not fully meet PLOS ONE’s publication criteria as it currently stands. Therefore, we invite you to submit a revised version of the manuscript that addresses the points raised during the review process.

We look forward to receiving your revised manuscript.

Kind regards,

Hadi Ghasemi

Academic Editor

PLOS ONE

Journal Requirements:

Reviewers' comments:

Reviewer's Responses to Questions

**Comments to the Author**

1. Does the manuscript provide a valid rationale for the proposed study, with clearly identified and justified research questions?

Reviewer #3: Yes

Reviewer #4: Yes

Reviewer #5: Yes

2. Is the protocol technically sound and planned in a manner that will lead to a meaningful outcome and allow testing the stated hypotheses?

Reviewer #3: Yes

Reviewer #4: Yes

Reviewer #5: No

3. Is the methodology feasible and described in sufficient detail to allow the work to be replicable?

Reviewer #3: Yes

Reviewer #4: Yes

Reviewer #5: Yes

4. Have the authors described where all data underlying the findings will be made available when the study is complete?

Reviewer #3: Yes

Reviewer #4: Yes

Reviewer #5: Yes

5. Is the manuscript presented in an intelligible fashion and written in standard English?

Reviewer #3: Yes

Reviewer #4: Yes

Reviewer #5: Yes

6. Review Comments to the Author

You may also provide optional suggestions and comments to authors that they might find helpful in planning their study.

Reviewer #3: Dear Authors,

no additional comments. All requested corrections have been entered in the text.

Thank you.

Best regards

Reviewer #4: Thanks to the precious authors for their great accuracy and interesting manuscript, many required items have been well added to the text of the manuscript.

Reviewer #5: This is an unblinded randomized noninferiority study of 5 mg pilocarpine vs. 2.5 mg pilocarpine plus herbal concoction. I have several statistical questions:

1. The randomization is stratified block but what are the stratification variables. A stratified randomization should be followed by a stratified analysis of the primary outcome, yet I see nothing on that.

2. Blocks of size 4. In an unblinded study, the investigator can determine the deterministic assignments in the tail of the block. That is why we use random block sizes.

3. "Opaque envelopes" were used in the 1948 streptomycin trial, but today we use a handy app to perform on site randomization in real time.

4. There is no discussion of why the trial must be unblinded. Surely the patients can take tablets without knowing what they are. How about 2 2.5 mg pills versus a 2.5 mg pill and herbal supplement? Why introduce bias in a study?

5. The final paragraph is in the past tense as though the trial has already occurred.

7. PLOS authors have the option to publish the peer review history of their article (what does this mean?). If published, this will include your full peer review and any attached files.

Reviewer #3: No

Reviewer #4: **Yes: **Fereshteh Najar-karimi

Reviewer #5: No

---

## [Author Response · Author response to Decision Letter 1]

29 Jun 2024

June 28, 2024

Dear Dr Hadi Ghasemi

Academic Editor

PLOS ONE

We wish to re-submit a revised manuscript titled “Efficacy of combined administration of Baekhogainsam-Tang and low-dose pilocarpine on frequent intractable xerostomia: Study protocol for a randomized controlled trial” by Su Il Kim et al., for consideration as a publication in the PLOS ONE. The manuscript number is PONE-D-24-16207R1.

We would like to thank editor and reviewers for their helpful comments and suggestions. We have revised our manuscript as per these comments. The following is a point-by-point response to the suggestions (bold) of the editor and reviewers (responses to the comments are provided in blue-colored text). The revised text in the manuscript has been highlighted in yellow.

Thank you very much for your consideration of our manuscript. We look forward to hearing from you.

Sincerely,

Young-Gyu Eun, MD, PhD

---

## [Decision Letter · Decision Letter 2]

9 Jul 2024

Efficacy of combined administration of Baekhogainsam-Tang and low-dose pilocarpine on frequent intractable xerostomia: Study protocol for a randomized controlled trial

PONE-D-24-16207R2

Dear Dr. Young-Gyu Eun,

We’re pleased to inform you that your manuscript has been judged scientifically suitable for publication and will be formally accepted for publication once it meets all outstanding technical requirements.

Kind regards,

Hadi Ghasemi

Academic Editor

PLOS ONE

Additional Editor Comments (optional):

Reviewers' comments:

Reviewer's Responses to Questions

**Comments to the Author**

1. Does the manuscript provide a valid rationale for the proposed study, with clearly identified and justified research questions?

Reviewer #5: Yes

2. Is the protocol technically sound and planned in a manner that will lead to a meaningful outcome and allow testing the stated hypotheses?

Reviewer #5: Yes

3. Is the methodology feasible and described in sufficient detail to allow the work to be replicable?

Reviewer #5: Yes

4. Have the authors described where all data underlying the findings will be made available when the study is complete?

Reviewer #5: Yes

5. Is the manuscript presented in an intelligible fashion and written in standard English?

Reviewer #5: Yes

6. Review Comments to the Author

You may also provide optional suggestions and comments to authors that they might find helpful in planning their study.

Reviewer #5: All comments have been addressed.xxxxxxxxxxxxxxxxxxxxxxxxxxxxxxxxxxxxxxxxxxxxxxxxxxxxxxxxxxxxxxxxxxx

7. PLOS authors have the option to publish the peer review history of their article (what does this mean?). If published, this will include your full peer review and any attached files.

Reviewer #5: No

---

## [Editor Report · Acceptance letter]

6 Aug 2024

PONE-D-24-16207R2 

PLOS ONE

Dear Dr. Eun, 

I'm pleased to inform you that your manuscript has been deemed suitable for publication in PLOS ONE. Congratulations! Your manuscript is now being handed over to our production team.

Kind regards, 

on behalf of

Dr. Hadi Ghasemi 

Academic Editor

PLOS ONE